# In Vitro Efficacy and Molecular Mechanism of Curcumin Analog in Pathological Regulation of Spinocerebellar Ataxia Type 3

**DOI:** 10.3390/antiox11071389

**Published:** 2022-07-18

**Authors:** Yu-Ling Wu, Jui-Chih Chang, Yi-Chun Chao, Hardy Chan, Mingli Hsieh, Chin-San Liu

**Affiliations:** 1Vascular and Genomic Center, Institute of ATP, Changhua Christian Hospital, Changhua 50091, Taiwan; ilenewu@gmail.com; 2Center of Regenerative Medicine and Tissue Repair, Changhua Christian Hospital, Changhua 50091, Taiwan; 145520@cch.org.tw; 3General Research Laboratory of Research Department, Changhua Christian Hospital, Changhua 50091, Taiwan; 4Inflammation Research & Drug Development Center, Changhua Christian Hospital, Changhua 50091, Taiwan; 183782@cch.org.tw; 5Allianz Pharmascience Limited, Taipei 10682, Taiwan; hwchan@rocketmail.com; 6Department of Life Science, Life Science Research Center, Tunghai University, Taichung 40704, Taiwan; mhsieh@thu.edu.tw; 7Department of Neurology, Changhua Christian Hospital, Changhua 50094, Taiwan; 8Graduate Institute of Integrated Medicine College of Chinese Medicine, China Medical University, Taichung 40447, Taiwan; 9Department of Post-Baccalaureate Medicine, College of Medicine, National Chung Hsing University, Taichung 40227, Taiwan

**Keywords:** spinocerebellar ataxia type 3, curcumin analog, nuclear factor erythroid-2 related factor 2, anti-oxidative enzymes, mitochondrial function

## Abstract

Unlike other nuclear factor erythroid-2-related factor 2 (Nrf2) activators, the mechanism of action of curcumin analog, ASC-JM17 (JM17), in regulating oxidative homeostasis remains unknown. Spinocerebellar ataxia type 3 (SCA3) is an inherited polyglutamine neurodegenerative disease caused mainly by polyglutamine neurotoxicity and oxidative stress. Presently, we compared actions of JM17 with those of known Nrf2 activators, omaveloxolone (RTA-408) and dimethyl fumarate (DMF), using human neuroblastoma SK-N-SH cells with stable transfection of full-length ataxin-3 protein with 78 CAG repeats (MJD78) to clarify the resulting pathological mechanism by assaying mitochondrial function, mutant ataxin-3 protein toxicity, and oxidative stress. JM17, 1 μM, comprehensively restored mitochondrial function, decreased mutant protein aggregates, and attenuated intracellular/mitochondrial reactive oxygen species (ROS) levels. Although JM17 induced dose-dependent Nrf2 activation, a low dose of JM17 (less than 5 μM) still had a better antioxidant ability compared to the other Nrf2 activators and specifically increased mitochondrial superoxide dismutase 2 in an Nrf2-dependent manner as shown by knockdown experiments with siRNA. It showed that activation of Nrf2 in response to ROS generated in mitochondria could play an import role in the benefit of JM17. This study presents the diversified regulation of JM17 in a pathological process and helped develop more effective therapeutic strategies for SCA3.

## 1. Introduction

Spinocerebellar ataxia type 3 (SCA3), also known as Machado–Joseph disease (MJD), is a rare autosomal dominantly inherited neurodegenerative disease [1,2] and is the most common SCA in Chinese and other Asian populations [3,4]. It belongs to the group of polyglutamine (polyQ) diseases which are caused by an abnormal expansion of cytosine–adenine–guanine (CAG) trinucleotides present in the coding regions of single, and otherwise unrelated, genes [5]. SCA3 is caused by 60–87 CAG repeats in the ATXN3 allele encoding a polyQ tract in the protein ataxin-3 and is expressed throughout the body and in neurons; however, mutated ataxin-3 protein causes neurotoxicity in specific brain regions in SCA3 patients, leading to progressive gait ataxia, dysarthria, dysphagia, muscular atrophy, and oculomotor dysfunction [6]. Although current preclinical data suggest that effective therapies are limited, aggregates of polyQ-expanded mutant protein are hallmarks of polyQ disease. A potential therapeutic strategy for polyQ disease involves combined induction of autophagy pathways and inhibition of oxidative stress, reducing mutant polyQ protein expression, polyQ aggregation, and neuron death [7].

Reactive oxygen species (ROS) are increased by oxidative stress, which is associated with pathogenesis of several late-onset polyQ neurodegenerative diseases [8]. Excessive ROS impairs mitochondrial respiratory function, consumes glutathione (GSH) content, and reduces antioxidant enzyme expression and activity in cells expressing full-length mutant ataxin-3 [9,10]. Our previous studies showed that SCA3 patients (with mutant ataxin-3 proteins) manifest diminished mitochondrial DNA copy numbers, reduced levels of antioxidants, and increased mitochondrial ROS [9,11]. Hence, improving mitochondrial function and antioxidant enzyme expression, or counteracting injurious effects of ROS, may contribute significantly to neuroprotection.

Nrf2 is a redox-sensitive transcription factor that protects against neurodegenerative diseases [12]. Studies of SCA3 disease have shown that polyQ mutant ataxin-3 decreases expression and transcription of Nrf2, contributing to impaired mitochondrial dynamics and increased oxidative stress [13]. Additionally, the Nrf2 pathway participates in autophagy induction, which helps to eliminate aggregation-prone proteins, but is impaired in neurodegenerative disorders, including SCA3 [14,15]. Aqueous extracts of Gardenia jasminoides (Rubiaceae) and Glycyrrhiza inflata (Fabaceae) upregulate anti-oxidative markers Nrf2, NAD(P)H quinone oxidoreductase 1 (NQO1), glutamate-cysteine ligase catalytic subunit, and glutathione S-transferase pi 1. They reduce production of ROS to diminish polyQ aggregation in cell models of SCA3 [10,16]. In Huntington’s disease models, mutant huntingtin (Htt) disrupts Nrf2 signaling, which contributes to impaired mitochondrial dynamics and may enhance susceptibility to oxidative stress in striatal cells [17]. Activated Nrf2 accumulates in nuclei and transcriptionally promotes expression of antioxidant response element (ARE)-mediated cyto-protective genes that encode antioxidant and phase 2 detoxifying proteins, such as heme oxygenase 1 (HO-1) and NQO1, glutathione reductase (GR), glutathione peroxidase (GPx), and glutamate-cysteine ligase (GCL) [18,19]. ARE-mediated genes reduce oxidative stress and prevent aggregation of mutant ataxin-3, thereby promoting cell survival [20,21]. Moreover, mitochondrial dysfunction contributes to neurodegenerative diseases, suggesting that phytochemicals may combat polyglutamine disease via improvements in mitochondrial function and the Nrf2 pathway [13,15,22]. These findings strongly suggest that activation of the Nrf2 pathway may be a viable treatment for SCA3.

Curcumin, derived from various species of Curcuma (Zingiberaceae), is known for its antioxidant, anti-inflammatory, and anticarcinogenic activities, and is thought to exert therapeutic effects on neurodegenerative disease by crossing the blood–brain barrier [23,24,25]. Allianz compound ASC-JM17 (JM17) (also called ALZ003) is a bioavailable curcumin analogue that degrades the polyQ androgen receptor (AR) via the ubiquitin–proteasome pathway and improves motor function in mouse models of spinal and bulbar muscular atrophy (SBMA), one of the polyQ diseases [26,27]. It is an FDA-approved AR inhibitor for treatment of SBMA [26], but the efficacy of SCA3 remains unclear. In order to demonstrate Nrf2 regulation in SCA3, small molecule Nrf2 activators, dimethyl fumarate (DMF) and omaveloxolone (RTA-408), are applied presently as the positive control treatments. DMF is the methyl ester of fumaric acid, mediates Nrf2 activation to act on Kelch-like ECH associating protein 1 (Keap1), and then drives the transcription of phase II antioxidant enzymes [28]. A 10 μM dose of DMF induces mitochondrial biogenesis primarily through the Nrf2 pathway in mice and human models of multiple sclerosis [29] and is able to attenuate ROS damage [30]. RTA-408 is a second generation semisynthetic oleanane triterpenoid compound. RTA-408 is an effective treatment for radiation-induced dermatitis, solid tumors of melanoma and lung cancer, Friedreich’s ataxia, and mitochondrial myopathies [31,32]. RTA-408 has a cytoprotective property by activating Nrf2 [33,34] and reduces human tumor cell growth at concentrations above 300 nM [33]. Thus, this paper investigates the mechanism of action of JM17 in improving mitochondrial activity, antioxidant levels, and reducing mutant ataxin-3 expression in an in vitro SCA3 model comprising human neural SK-N-SH cells expressing mutated ataxin-3 with 78 glutamine residues, which causes SCA3 pathology. Moreover, through comparison of the effects conducted by small molecule Nrf2 activators, DMF and RTA-408, the regulatory role of JM17 in SCA3 is more clearly defined.

## 2. Materials and Methods

### 2.1. Materials

A wild-type human neuroblastoma cell line (SK-N-SH) was provided by Prof. Mingli Hsieh (Department of Life Science, Tunghai University, Taiwan). SK-N-SH cells expressing 26 (MJD26) or 78 glutamines (MJD78) in full-length ataxin-3 constructs containing 26 or 78 CAG repeats have been reported previously [35,36]. 3-Methyladenine (3MA) was purchased from Sigma Chemical Co. (St. Louis, MO, USA). G418 was obtained from InvivoGen (San Diego, CA, USA). JM17, DMF, and RTA-408 were provided by Allianz Pharmascience Limited (Taipei, Taiwan). Specific primary antibodies included p62, ataxin-3, Lamin B (1:1000 dilution, ab56416, ab175265, ab16048; Abcam, Cambridge, UK), HO-1 (1:1000 dilution, 610712; BD Biosciences, Boston, MA, USA), NOQ-1, Nrf2 (1:1000 dilution, sc-376023, sc-722; Santa Cruz Biotechnology, CA, USA), superoxide dismutase (SOD)1, SOD2 (1:1000 dilution, GTX100554, GTX116093; Genetex Inc., Irvine, CA, USA), microtubule-associated protein 1 light chain 3 (LC3) (1:1000 dilution, 4108S; Cell Signaling Technology Inc., Beverly, MA, USA), and glyceraldehyde 3-phosphate dehydrogenase (GAPDH) (1:1000 dilution, MAB374; Millipore, Billerica, CA, USA). Nuclear Extraction Kits and PROTEOSTAT^®^ Protein aggregation assays were from Abcam Inc. (ab113474; Cambridge, MA, USA) and Enzo Life Sciences (Farmingdale, NY, USA), respectively.

### 2.2. Cell Culture and Treatment

SK-N-SH WT, MJD26, and MJD78 cells were grown in Dulbecco’s modified Eagle’s medium (DMEM) (high glucose, DMEM-HG) (GIBCO/Invitrogen, Carlsbad, CA, USA) containing 10% heat-inactivated fetal bovine serum (HyClone Laboratories, Logan, UT, USA), 1% L-glutamine (GIBCO), 1% non-essential amino acids (GIBCO), and 1% penicillin/streptomycin (GIBCO). Culture medium was supplemented with 0.1 mg/mL G418 (Invivogen, San Diego, CA, USA) to maintain stable exogenous protein expression in MJD26 and MJD78 cells. The medium was changed every 2 days and different types of cells were subcultivated weekly at a ratio of 1:2 and maintained at 37 °C in a humidified atmosphere with 5% CO2. MJD26 and MJD78 cells were used with a passage number of 39–45 in each experiment. Culture medium supplemented with 100 µg/mL G418 (InvivoGen, San Diego, CA, USA) was used to maintain exogenous protein expression in MJD26 and MJD78 cells. MJD78 cells were exposed to medium containing JM17 in doses of 0.3, 1, and 5 μM, or 0.3 μM RTA-408, or 10 μM DMF for 24 h. Simultaneously, cells treated with dimethyl sulfoxide (DMSO) at a concentration of 0.025% (the same concentration used for the highest dosage of JM17) were incubated under the same conditions used for the vehicle control. After treatments, cells were collected for the drug safety evaluation using a cell viability assay and the drug’s effectiveness was validated by the mitochondrial functions, the levels of mutant ataxin-3, and oxidative stress, respectively.

### 2.3. Cell Viability Assay

Cell viability was assessed by propidium iodide (PI) (Invitrogen) uptake using a flow cytometer (BD Biosciences, Franklin Lakes, NJ, USA). Treated cells were rinsed with PBS and incubated with 10 µg/mL PI in PBS at 37 °C for 15 min. Cell survival was assessed as the percentage of dead (PI-positive) cells.

### 2.4. Mitochondrial Respiration

Mitochondrial respiration rates were measured in SK-N-SH WT, MJD26, and MJD78 cells at 37 °C with a high-resolution Oroboros Oxygraph 2K respirometer (Oroboros, Oxygraph; Innsbruck, Austria) according to the Buck et al. study [37]. O_2_ flux served as a surrogate for mitochondrial electron transport chain performance. O_2_ flux was measured at baseline and after progressive titration with 0.5 mM malate, 10 mM L-glutamate, 2.5 mM adenosine diphosphate (ADP), 10 mM succinate, 5 µM oligomycin, 10 µM carbonyl cyanide-p-(trifluoromethoxy) phenylhydrazone (FCCP), 0.5 µM rotenone, and 5 µM antimycin A. After addition of malate, L-glutamate, and ADP, O_2_ flux was used to measure mitochondrial respiratory complex I activity. The basal respiration rate of the mitochondrial electron transport chain was determined after administration of the complex I substrate, ADP. The adenosine triphosphate (ATP) synthase inhibitor, oligomycin, was added to remove the pH gradient, enabling maximal rates of electron transport and ATP-linked respiration to occur. A combination of rotenone and antimycin A was employed to block respiratory electron flux at mitochondrial complexes I and III. Spare respiratory capacity was calculated from the difference between the maximal respiration achieved with uncoupled FCCP and basal respiration. Maximum uncoupled capacity was defined as the ratio of the uncoupled respiration and oligomycin-treated respiration rates.

### 2.5. RNA Extraction and Quantitative Real-Time RT–PCR Analysis

Total RNA was extracted from cells by using the TRIzol reagent (Invitrogen). Total RNA was reverse transcribed with the NucleoSpin RNA II Kit (Macherey-Nagel, Düren, Germany) and reverse transcribed using the Transcriptor First Strand cDNA Synthesis kit (Roche Applied Science, Indianapolis, IN, USA) for synthesis of complementary DNA. The expressions of mRNA were amplified by complementary DNA with the SYBR Green PCR Master Mix (Roche Applied Science) and determined by quantitative analysis of real-time RT–PCR using an ABI Prism 7300 system (Applied Biosystems, Waltham, MA, USA). The primers targeted at non-CAG-repeated regions of *ATXN3* (Forward: 5′-AAGAGACGAGAAGCCTAC-3′ and Reverse: 5′-TTCACTCATAGCATCACCTA-3′) and *β-actin* (Forward: 5′-ATCGTGCGTGACATTAAGAGAAG-3′ and Reverse: 5′-AGGAAGGAAGGCTGGAAGAGTG-3′) used for RT–PCR were applied. Real-time RT–PCR was performed as follows: 1 cycle of hot start at 95 °C for 10 min and 40 cycles of 15 s denaturation at 95 °C then annealing and extension at 60 °C for 90 s. The mRNA expressions normalized to β-actin were respectively presented as relative expression levels.

### 2.6. Measurements of GSH Content

After treatment for 24 h, cells were collected on iced PBS. GSH content in MJD26 and MJD78 cells was measured using a GSH + GSSG/GSH assay kit (abcam ab239709; Cambridge, MA, USA) according to the manufacturer’s instructions. Colorimetric intensity at 405 nm was measured using a FCLARIOstar Plate Reader (BMG LabTech, Ortenberg, Germany).

### 2.7. Measurements of Catalase Activity

After treatment for 24 h, cells were collected on iced PBS. Catalase activity was assayed in MJD26 and MJD78 cells using AmpliteTM Fluorimetric Catalase assay kits (AAT Bioquest 11306, Sunnyvale, CA, USA) according to the manufacturer’s instructions. Fluorescence intensity (Ex/Em 540/590 nm) was measured using a FCLARIOstar Plate Reader (BMG LabTech, Ortenberg, Germany).

### 2.8. Protein Extraction and Western Blots

Total protein extracts were prepared from treated cells in RIPA buffer (50 mM Tris-Cl pH 7.4, 150 mM NaCl, 1% NP40, 0.25% Na-deoxycholate, 1 mM PMSF) (Cat.87787; Thermo Fisher Scientific Inc. Carlsbad, CA, USA) containing a protease inhibitor cocktail (Sigma-Aldrich, St. Louis, MO, USA), incubated on ice for 30 min, and then spun at 14,000× *g* for 10 min at 4 °C. Nuclear protein extracts were prepared by using the nuclear extraction kit according to the manufacturer’s instructions. Whole-cell lysates and nuclear fractions were quantified using a BCA assay (Pierce, Rockford, IL, USA). Equal amounts of proteins were fractionated and separated electrophoretically on 10% or 12% SDS-polyacrylamide gels (Bio-Rad Laboratories, Richmond, CA, USA) and blotted onto polyvinylidene difluoride membranes (Merck KGaA, Darmstadt, Germany). Non-specific binding was blocked with BlockPROTM Protein Blocking Buffer (Visual Protein Biotech., Taipei, Taiwan). Membranes were blotted with specific antibodies and horseradish peroxidase-conjugated anti-mouse or anti-rabbit IgGs (Jackson Immunoresearch Lab, West Grove, PA, USA). Protein intensity was determined with an enhanced chemiluminescence reagent (Immobilon Western, Millipore, Billerica, CA, USA) and images were captured with a Fusion FX7 system (Vilber Lourmat, Marne-la-Vallée, Collégien, France).

### 2.9. Measurements of Mitochondrial ROS, Total ROS, and Protein Aggregation

Levels of mitochondrial ROS, total ROS, and protein aggregation in MJD26 and MJD78 cells were determined using MitoSOX Red reagent (M36008, Invitrogen), CellROX™ Orange Flow Cytometry Assay Kits (C10493, Invitrogen), and PROTEOSTAT^®^ Protein aggregation assays, respectively. After treatment for 24 h, cells were washed with PBS, trypsinized with 0.05% trypsin, and collected into PBS. After treating 1 × 10^5^ cells/mL with 5 μM MitoSOX Red and 500 nM CellROX™ Orange reagent, cells were incubated for 10 min at 37 °C while protected from light. Cells were washed gently 3× with PBS. Fluorescence intensity was measured using a FACSCalibur flow cytometer (BD Biosciences, Franklin Lakes, NJ, USA) and an FCLARIOstar Plate Reader (BMG LabTech, Ortenberg, Germany).

### 2.10. Plasmids, Transfection, and Reporter Gene Assay

The pGL3 promoter-luciferase plasmids expressing a 2xARE fragment containing two tandem repeats of double-stranded oligonucleotides spanning the Nrf2 binding site, 5′-TGACTCAGCA-3′, were used to measure Nrf2 transcriptional activity [38]. A pGL3-2xARE/Luc vector was provided by Prof. Haw-Wen Chen (Department of Nutrition, China Medical University, Taiwan). Cells were transiently transfected for 16 h with 0.1 μg of the pGL3-2xARE/Luc vectors, as well as 0.3 μg of the pSV-β-galactosidase control vector, by use of Lipofectamine 2000 transfection reagent (Invitrogen, Waltham, MA, USA) and were then treated with JM17, RTA-408, or DMF for 24 h. The Nrf2 transcriptional luciferase activity was measured using the Luciferase Assay System (Promega, Madison, WI, USA) and was corrected on the basis of β-galactosidase activity.

### 2.11. Nrf2 Small Interfering RNA (siRNA) Transfection

To knockdown Nrf2 expression, MJD78 cells were transfected with an siRNA against Nrf2 (ID: s9492) (6 nM) and non-targeting control (NTC) siRNA (Ambion Life Technologies Invitrogen, Carlsbad, CA, USA) using Lipofectamine RNAiMAX transfection reagent according to the manufacturer’s instructions. After 16 h of transfection, cells were treated with JM17, RTA-408, or DMF, as described in figure legends.

### 2.12. Statistical Analysis

All experiments were performed at least in triplicate. Data are presented as means ± SD. Comparisons of differences among multiple experimental conditions were evaluated using one-way ANOVA followed by post hoc Tukey’s multiple comparison test (SPSS 17.0 Statistics, SPSS Inc., Chicago, IL, USA). *p* < 0.05 was considered statistically significant.

## 3. Results

### 3.1. Functional Support of Cell Viability and Mitochondrial Function

To explore the potential of JM17, RTA-408, and DMF for treating SCA3, we used wild-type SK-N-SH neuroblastoma cells (WT) and SK-N-SH cells containing either 26 or 78 glutamines in the ataxin-3 gene, termed MJD26 and MJD78 cells hereafter. Incubation of WT, MJD26, and MJD78 cells with 0.3, 1, or 5 μM JM17, 0.3 μM RTA-408, or 10 μM DMF resulted in no cytotoxicity for 24 h (Figure 1A and Appendix A). Even RTA-408 (≤1 μM) and DMF (≤20 μM) had no effect on the viability of MJD78 cells for 24 h (Appendix A); the doses of 0.3 μM RTA-408 or 10 μM DMF were selected based on the relatively better inhibitory effect of total ROS production which is shown in Appendix A. Moreover, JM17 > 5 μM increased cytotoxicity for 24 h in the WT and MJD78 cells (Appendix A). Thus, according to those findings, we used doses of 0.3, 1, or 5 μM JM17, 0.3 μM RTA-408, and 10 μM DMF for 24 h to perform the following experiment. Oxygen consumption (O_2_ flux) was measured to evaluate overall mitochondrial respiration in WT, MJD26, and MJD78 cells using the high-resolution Oroboros Oxygraph 2K respirometer (Appendix A) referred to our previous study [36]. Quantification of oxygen consumption, including basal respiration, ATP-linked respiration, and maximum uncoupled capacity, was significantly reduced in MJD78 cells compared to WT and MJD26 cells. In MJD78 cells, basal respiration increased in all treatments (6.2× in 0.3 µM JM17; 5.6× in 1 μM JM17; 7.7× in 0.3 μM RTA-408; and 6.3× in 10 μM DMF). Basal respiration in 5 μM JM17 was the same as the control; thus, it appears that this concentration is ineffective in MJD78 cells (Figure 1B). To gain further insight into production of ATP by JM17, we investigated ATP-linked respiration in MJD78 cells. Using 1 μM JM17 (4.43 ± 0.76 pmol/million cells, *p* = 0.0012) and 10 μM DMF (3.68 ± 0.53 pmol/million cells, *p* = 0.0007) not only significantly increased ATP production, but also completely restored it to levels seen in MJD26 cells. Moreover, ATP-linked respiration in 0.3 μM JM17 was less augmented than 1 μM JM17 and responded similarly to 5 μM JM17 or control (Figure 1C). The maximum uncoupled respiration state was attained by adding FCCP, revealing excess expression in MJD78 cells with 0.3 (11.76 ± 1.0 pmol/million cells, *p* = 0.0034) and 1 μM (9.72 ± 1.31 pmol/million cells, *p* = 0.039) JM17, 0.3 μM RTA-408 (14.06 ± 2.05 pmol/million cells, *p* = 0.0057), and 10 μM DMF (11.3 ± 0.97 pmol/million cells, *p* = 0.0047) treatment, as compared to MJD26 cells (Figure 1D). These data showed that treatment with 1 μM JM17 drastically improved dysfunctional mitochondrial respiration of MJD78 cells and has an effect similar to that of 0.3 μM RTA-408 or 10 μM DMF.

### 3.2. Induction of Autophagy-Mediated Degradation of PolyQ Mutant Ataxin-3 Expression and Decrease in Protein Aggregation

As in our previous study [9], MJD78 cells showed increases in mutant ataxin-3 protein expression and protein aggregation compared to MJD26 cells. We found that JM17, RTA-408, or DMF exposure has no significant effect on ATXN3 mRNA level in MJD78 cells (Figure 2A). Using immunoblotting and fluorescence intensity, respectively, we showed that incubation of MJD78 cells with 0.3 or 1 μM JM17, 0.3 μM RTA-408, or 10 μM DMF treatment for 24 h significantly decreased protein aggregation (Figure 2B: 0.76 ± 0.02× in 0.3 µM JM17, *p* < 0.0001; 0.65 ± 0.08× in 1 μM JM17, *p* = 0.0015; 0.57 ± 0.08× in 0.3 μM RTA-408, *p* = 0.0007; and 0.71 ± 0.08× in 10 μM DMF, *p* = 0.0035) and mutant ataxin-3 protein expression (Figure 2C: 0.76 ± 0.04× in 0.3 µM JM17, *p* < 0.0001; 0.7 ± 0.07× in 1 μM JM17, *p* < 0.0001; 0.33 ± 0.03× in 0.3 μM RTA-408, *p* < 0.0001; and 0.84 ± 0.05× in 10 μM DMF, *p* = 0.0005). Therefore, since activation of autophagy could attenuate mutant ataxin-3 protein and aggregation [39], we investigated whether JM17 promotes autophagy. Western blotting confirmed that 0.3 or 1 μM JM17 significantly upregulated expression of p62 and LC3 II in MJD78 cells (Figure 2D). Moreover, the capability of 1 μM JM17 or 0.3 μM RTA-408 to reduce protein aggregation and mutant ataxin-3 protein expression in MJD78 cells was annulled by pretreatment with 3MA, an autophagy inhibitor (Figure 2E–G). These data show that JM17 and RTA-408 may be responsible, at least in part, for degradation of protein aggregates by induction of autophagy.

### 3.3. Reduction in Oxidative Stress including ROS Formation and Antioxidant Enzyme Activities

Excessive mitochondrial ROS production contributes to mitochondrial damage and cell death and is involved in pathogenesis of diabetes, Alzheimer’s disease, and Parkinson’s disease [40,41]. Additionally, compared to WT and MJD26 cells, mitochondrial and total ROS levels were increased in MJD78 cells, as determined using MitoSOX and CellROX™ Orange fluorescence [39]. Mitochondrial and total ROS were upregulated in the DMSO vehicle control, whereas under 0.3 or 1 μM JM17, in contrast to under RTA-408 and DMF treatment, mitochondrial and total ROS levels were more effectively reduced in MJD78 cells (Figure 3A,B). Moreover, our present data revealed that the antioxidant system was damaged in MJD78 cells, as evidenced by decreased protein expression of HO-1, NQO1, SOD1, and SOD2 (Figure 3C), as well as total GSH, reduced GSH, and catalase levels (Figure 3D–F). Supplementation with 1 μM JM17 dramatically increased NQO1 (3.18 ± 0.34×, *p* = 0.004), HO-1 (2.31 ± 0.13×, *p* < 0.0001) and SOD2 (2.48×, *p* < 0.0001) protein expression (Figure 3C) and catalase levels (Figure 3F, 1.49 ± 0.15×, *p* < 0.0001), as well as both total and reduced GSH levels (Figure 3D, total GSH: 6.73 ± 0.92×, *p* < 0.0001; Figure 3E, reduced GSH: 4.68 ± 0.41×, *p* < 0.0001) in MJD78 cells. RTA-408 increased NQO1, HO-1, SOD1, SOD2, and catalase, total and reduced GSH levels less effectively than 1 μM JM17 (Figure 3C–F). Taken together, treatment with 1 μM JM17 and RTA-408 not only reduced ROS production but also renewed antioxidant enzyme protein expression in MJD78 cells.

### 3.4. Nrf2 Activation-Dependent Regulation in Raising Antioxidation and Decreasing Mutant Ataxin-3 Expression

Nrf2 is a critical transcription factor that serves an antioxidant protective function to minimize oxidative stress in neurodegenerative diseases [12]. Our data reveal that nuclear and cytosol Nrf2 protein expression and transcriptional activity were reduced in MJD78 cells compared to MJD26 cells (Figure 4A,B). Nrf2 transcriptional activity was significantly enhanced in MJD78 cells after JM17 or RTA-408 treatment (Figure 4B: 1.68 ± 0.08× in 0.3 µM JM17, *p* = 0.0001; 3.51 ± 0.13× in 1 μM JM17, *p* < 0.0001; 4.05 ± 0.06× in 5 μM JM17, *p* < 0.0001; 3.61 ± 0.14× in 0.3 μM RTA-408, *p* < 0.0001; and 1.71 ± 0.1× in 10 μM DMF, *p* = 0.0003). To confirm whether JM17 and RTA-408 can induce Nrf2 activation, we used Nrf2 siRNA to measure Nrf2 transcriptional activity and to target antioxidant protein expression in MJD78 cells. Western blot and luciferase analysis showed that Nrf2 siRNA also decreased Nrf2 protein expression and ARE luciferase activity of its downstream genes, NQO1 and SOD2, compared to Nrf2 NTC (Figure 5A,B). In contrast, Nrf2 transcriptional activity and antioxidation enhanced by JM17 and RTA-408 treatment were strongly reversed by Nrf2 siRNA in MJD78 cells, confirming that JM17 and RTA-408 induce antioxidant effects in MJD78 cells through Nrf2 activation (Figure 5A–E). Interestingly, only RTA-408 restored mutant ataxin-3 expression in MJD78 cells by silencing Nrf2 (Figure 5A). The data revealed the machinery of RTA-408 was more reliant on the autophagy-activated clearance by Nrf2 in contrast to JM17 with greater anti-oxidation performance.

## 4. Discussion

In this study, we showed that appropriate concentrations of JM17 (0.3 and 1 μM), RTA (0.3 μM), and DMF (10 μM) improved mitochondrial function and promoted autophagy, thereby facilitating degradation of mutant ataxin-3 to reduce neurotoxic accumulations of misfolded protein aggregates. Notably, JM17 also has relatively good antioxidant activity compared to RTA and DMF, as evidenced by its capacity to significantly reduce total and mitochondrial ROS, increase antioxidant enzymes (especially HO-1 and SOD2), and increase total/reduced GSH. Moreover, SOD2, a key mitochondrial antioxidant enzyme, is only induced by JM17 and not by RTA or DMF. On the other hand, intriguingly, the benefits of JM17 are not mainly obtained from initiation of Nrf2 activation due to effective doses of 0.3 and 1 μM, with a lower induction of Nrf2 activation in contrast to the 5 μM dose which was an ineffective dose of JM17. The doses above 10 μM significantly decreased cell viability whenever in WT or MJD78 cells (Appendix A). JM17 has recently been shown to increase expression of proteasome subunits, antioxidant enzymes, and molecular chaperones via actin or the Nrf1/Nrf2 pathway to mitigate toxicity of the mutant androgen receptor responsible for SBMA in cell, fly, and mouse models [26]; however, the detailed mechanism of action in neurodegenerative diseases such as SCA3 remains unknown. Our data showed that, compared to the 0.3 μM JM17 treatment, mild elevation of mitochondrial ROS (not total ROS) concomitant with an increase in mitochondrial antioxidant enzyme (SOD2) actually contributes to not only facilitating the elimination of toxic mutant ataxin-3 protein but also significantly decreasing protein aggregates in 1 μM dose-exposed disease cells. The mitochondrial function also becomes better. Our data showed that the expression of normal ataxin-3 protein outperformed that of mutant ataxin-3 protein (Figure 2C). We found that endogenous ataxin-3 protein compensatively augmented cell survival against stress from the gradually increasing mutant ataxin-3 protein performance with a growing number of cell passages [36]. Moreover, the ubiquitin–proteasome activity of ataxin-3 is not changed by the presence of the polyglutamine-expanded mutant protein [42], its clearance predominantly dependent on autophagy with autophagosome lysosome to degrade [43,44]. In contrast, JM17 and RTA-408 reduced mutant ataxin-3 protein rather than its mRNA level in MJD78 cells because the protein’s half-life results in an increased rate of autophagic degradation (Figure 2A,C) with the decreased protein numbers not observably compensated by increasing the transcriptional and translational rates [36]. Interestingly, our data reflected that the JM17-mediated decrease in mutant ataxin-3 was not affected by silencing Nrf2, contrary to RTA-408 treatment (Figure 5A). This might be, in part, due to autophagic induction and subsequent degradation of mutant ataxin-3 protein in an Nrf2-dependent manner [39]. Curcumin mitigates α-synuclein via the GSK-3β inhibition-mediated autophagy/lysosomal pathway in human neuroblastoma cells [45]. It is possible that JM17 (curcumin analogue) produced mutant ataxin-3 cleaning via autophagy with the Nrf2-independent pathway. Thus, we suggest that JM17-mediated regulation of mitochondrial redox homeostasis could play a role in controlling cell or mitochondrial quality through the process of engaging autophagy [46], which does not seem to correlate with cell toxicity [9]. However, this assumption still needs to be proven in future studies. Moreover, curcumin is also found to inhibit the oligomerization of tau and disintegrated preformed tau filaments in Alzheimer’s disease [47]. Whether there is similar machinery implied in the regulation of JM17 in SCA3 or in the other polyQ diseases deserves further studies.

The present findings further support the above study by confirming that JM17-mediated activation of the Nrf1/Nrf2 pathway is a viable option for pharmacological intervention not only for SBMA, but also for other polyglutamine diseases. Interestingly, the latest study indicates that a JM17 concentration greater than or equal to 2 μM enhances accumulation of ROS, augments lipid peroxidation, and blocks GPx to suppress growth of temozolomide (TMZ)-sensitive glioblastoma through JM17-mediated androgen receptor ubiquitination; however, a decrease in cell viability occurred in normal astrocytes at more than a 10 μM dose of JM17 [27], which is similar to our finding in SK-N-SH neurons of WT and MJD78 where obvious cytotoxicity was observed at concentrations above 10  μM (Appendix A). Moreover, JM17-deprived cell viability at high dose treatments (5 μM and 10 μM) was not proportional to the dramatic increase in ROS in TMZ-sensitive glioblastoma [27]. This echoes our findings with different therapeutic responses in effective dosages of JM17 (as described later) and the implication of the critical role that JM17 concentration performs in the regulatory process of pathologic mechanisms. Thus, we suggest that JM17 regulates oxidative stress and cell functions by indirect and diverse mechanisms determined by different disease types.

Nrf2 is a transcription factor well-known for controlling basal and inducible expression of various antioxidant and detoxifying enzymes. It can be activated through exercise [48], calorie restriction (including fasting) [49], and natural (e.g., curcumin and resveratrol) [50] and chemical activators (e.g., DMF, RTA-408) [51]. Activators differ in Nrf2 induction efficiency and downstream functional consequences [52]. To date, the most successful Nrf2 activator is DMF, which has been approved for the clinical treatment of relapsing-remitting multiple sclerosis [53] and for treatment of psoriasis [54]. Another novel activator, RTA-408, is currently in phase II clinical trials for Friedreich’s ataxia [55], as well as for treatment of ocular inflammation and pain after ocular surgery [55] and diabetic wound recovery [56]. In this study, we initially compared therapeutic effects and mechanisms of JM17, DMF, and RTA-408 on SCA3. All of these activators exerted a therapeutic effect in an in vitro model of SCA3 at therapeutically relevant doses, including improvements in mitochondrial function, degradation of mutant ataxia-3 protein, and reduction in oxidative stress. Notably, JM17 inhibited total and mitochondrial ROS more effectively than the other two and simultaneously increased the level of SOD2, which the other two activators did not do. DMF was as good as JM17 at increasing ATP-linked respiration, but it was less effective than RTA-408 and JM17 in degrading mutant ataxia-3 protein via autophagy. These results illustrate the diverse regulation of Nrf2 activators mentioned above, and mechanisms of individual Nrf2 activators in SCA3 may still differ more than we currently realize. Nrf2 activation has been recognized as a comprehensive strategy for the treatment of neurodegenerative disorders due to the capacity of Nrf2 to regulate intermediary metabolism and mitochondrial function [57]. It is well known that Nrf2 activators have multiple targets due to their electrophilicity and interference with protein–protein interactions [58]. It will be worth further clarifying differences among Nrf2 activators in the regulation of mitochondria, such as mitochondrial dynamics, homeostasis, and mitophagy.

These results showed a JM17-mediated increase in Nrf2 nuclear translocation in a dose-dependent manner to induce downstream gene encoding of antioxidant enzymes (except SOD2). However, therapeutic responses, including improvement of mitochondrial function, reduced oxidative stress, and increased mutant protein degradation, did not increase with dose. Moreover, induction was consistently abolished at higher doses (5 μM). This result demonstrates that activation of antioxidant Nrf2 signaling was not the only important mechanism in this process, and we suggest that mitochondrial regulation could also be involved because the mitochondrial antioxidant enzyme, SOD2, was only induced by JM17, which has a higher antioxidant capacity than RTA-408 and DMF. Furthermore, similar to other antioxidant enzymes, induction of SOD2 still requires activation via Nrf2 because transient Nrf2 silencing reduced the level of SOD2. We are currently trying to determine whether the regulatory effect of JM17 on mitochondrial metabolism or mitochondria-related pathways depends on the Nrf2 pathway. In our preliminary results, JM17 distribution was observed not only in nuclei, but also in mitochondria (Appendix A). Thus, the possibility that JM17 may regulate mitochondrial function through Nrf2-independent pathways cannot be excluded but requires further confirmation with a comprehensive analysis of mitochondrial dynamics, mitophagy, and homeostasis, etc. Possible mechanisms of JM17 on the Nrf2 pathway, clearance of protein aggregation, and level of oxidative stress are illustrated in Figure 6.

Mitochondrial targeting therapy is a potential therapeutic strategy for SCA3. In vitro and in vivo studies have demonstrated that recovery of mitochondrial function with phototherapy can decrease mutant protein accumulation, protect against Purkinje cell death, and slow progression of SCA3 [36,59]. Nrf2 activation regulates mitochondrial function in multiple directions and exhibits diverse effects in different kinds of cells [57,60]. This study showed that variation in Nrf2-mediated regulation of cellular machinery and function in SCA3 cells depends on the types and doses of activators. Unlike other Nrf2 activators, JM17 increases the performance of mitochondrial antioxidant enzymes, which may explain why it is currently the most promising antioxidant for SCA3 treatment.

## Figures and Tables

**Figure 1 antioxidants-11-01389-f001:**
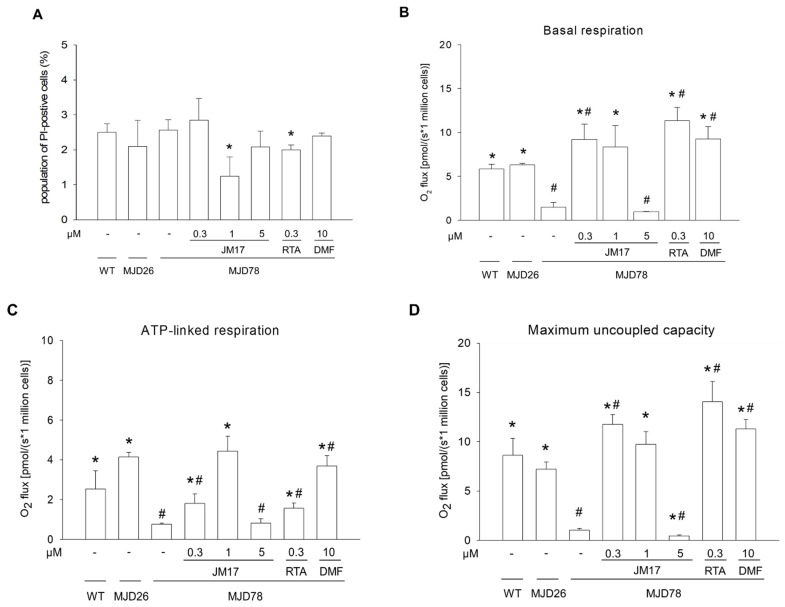
Functional support of cell viability and mitochondrial function. Cells were treated with or without DMSO vehicle control or with 0.3, 1, or 5 μM JM17, 0.3 μM RTA-408, or 10 μM DMF for 24 h. (**A**) Cell viability was assessed by flow cytometry analysis using propidium iodide to stain dead cells. (**B**) Basal respiration, (**C**) ATP-linked respiration, and (**D**) Maximum uncoupled capacity were quantified with an OroborosR Oxygraph-2K assay. Data are presented as the mean ±SD of at least three independent experiments and are expressed as multiples of values. * *p* < 0.05, compared to non-treated MJD78 group; ^#^ *p* < 0.05, compared to MJD26 group.

**Figure 2 antioxidants-11-01389-f002:**
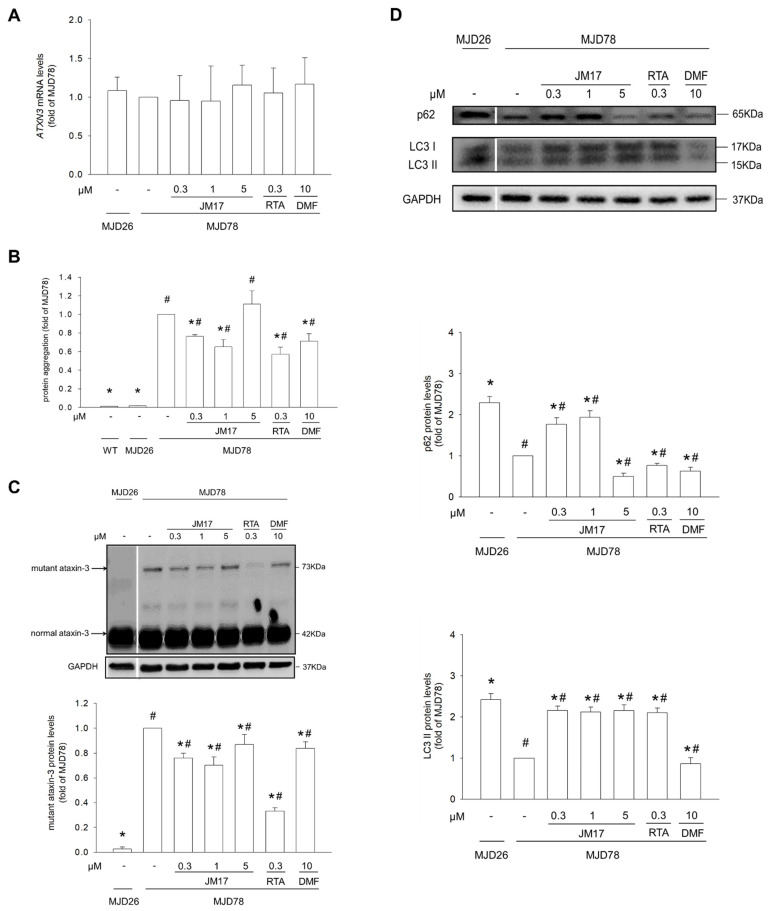
Induction of autophagy-mediated degradation of polyQ mutant ataxin-3 expression and decrease in protein aggregation. Cells were treated with or without DMSO vehicle control or with 0.3, 1, or 5 μM JM17, 0.3 μM RTA-408, or 10 μM DMF for 24 h. (**A**) The mRNA level of *ATXN3* was detected by using quantitative real-time RT–PCR analysis. The *ATXN3* mRNA expression was quantified by normalized β-actin. (**B**) Protein aggregation was assessed using the PROTEOSTAT^®^ protein aggregation assay. (**C**) Protein expression of mutant and normal ataxin3. (**D**) Protein expression of p62 and LC3-II. Cells were treated with or without DMSO vehicle control or with 1 μM JM17 or 0.3 μM RTA-408 for 24 h in the absence or presence of pretreatment with 1 mM 3MA for 1 h. (**E**) Protein expression of p62 and LC3-II, (**F**) protein aggregation, and (**G**) protein expression of mutant and normal ataxin3 in 3MA treatment. Data are presented as the mean ±SD of at least three independent experiments. Values from the treated cells were normalized to those of the MJD78 cells treated with vehicle only. In 3MA treatment graphs, within treatments with the same absence or presence of 3MA, values are expressed as the percentage of MJD78 cells treated with vehicle only. * *p* < 0.05, compared to non-treated MJD78 group; ^#^ *p* < 0.05, compared to MJD26 group; ^+^ *p* < 0.05, compared to non-treated MJD78 with absence of 3MA group.

**Figure 3 antioxidants-11-01389-f003:**
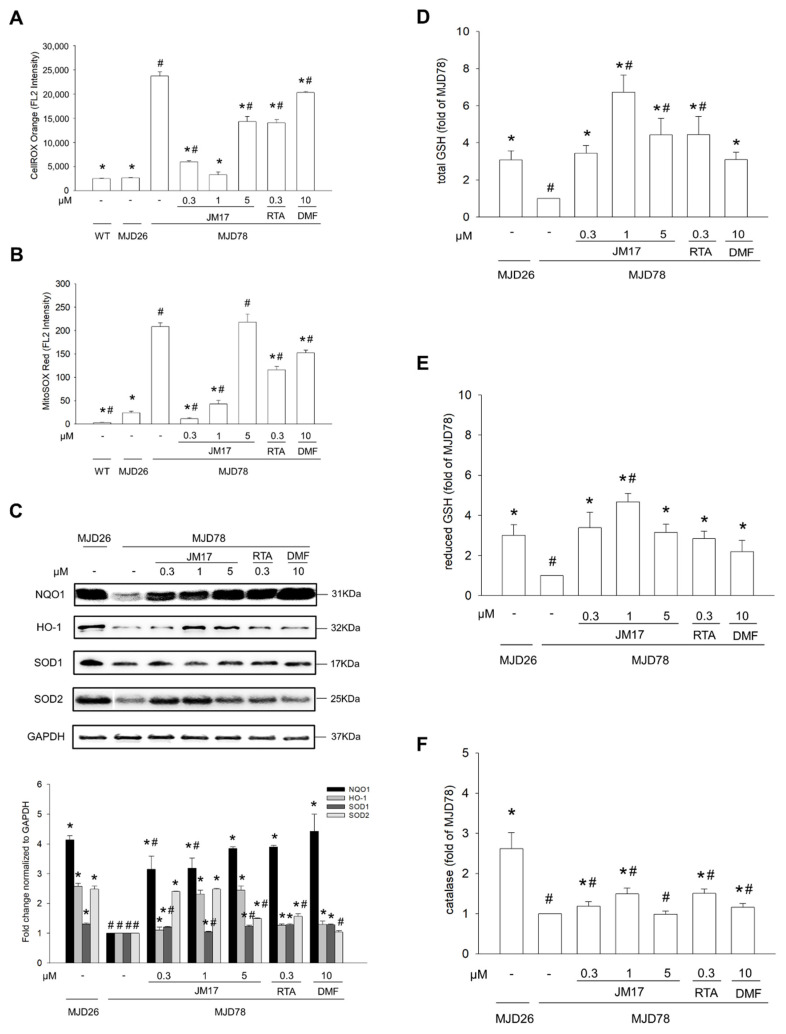
Reduction in oxidative stress including ROS formation and antioxidant enzyme activities. Cells were treated with or without DMSO vehicle or with 0.3, 1, or 5 μM JM17, 0.3 μM RTA-408, or 10 μM DMF for 24 h. (**A**) Total and (**B**) mitochondrial ROS were measured with CellROX Orange and MitoSOX Red staining, respectively, and were quantified by flow cytometry. (**C**) Protein expression of NQO1, HO-1, SOD1, and SOD2. (**D**) Total GSH, (**E**) reduced GSH levels, and (**F**) catalase activity were measured with a GSH + GSSG/GSH assay kit and an AmpliteTM Fluorimetric Catalase assay kit, respectively, and were quantified using a FCLARIOstar Plate Reader. Data are presented as the mean ±SD of at least three independent experiments. Data are expressed as multiples of values in (**A**,**B**) graphs and values from the treated cells were normalized to those of the MJD78 cells treated with DMSO vehicle only in graphs (**C**–**F**). * *p* < 0.05, compared to non-treated MJD78 group; ^#^ *p* < 0.05, compared to MJD26 group.

**Figure 4 antioxidants-11-01389-f004:**
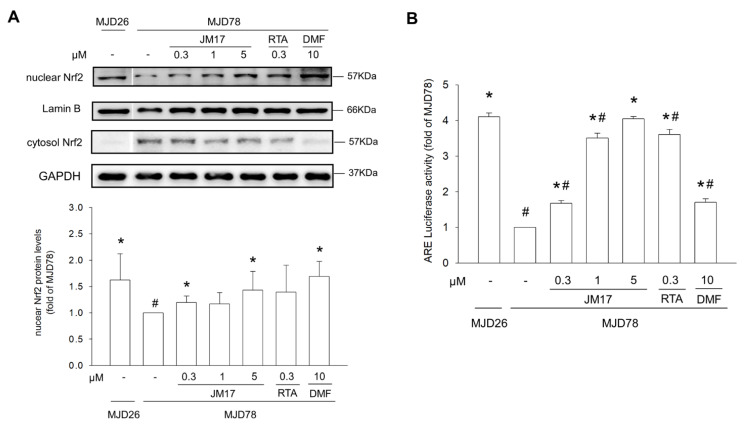
Effects of JM17, RTA-408, and DMF on Nrf2 activation in MJD78 cells. Cells were treated with or without DMSO vehicle control or with 0.3, 1, or 5 μM JM17, 0.3 μM RTA-408, or 10 μM DMF for 24 h. (**A**) Protein expression of nuclear and cytosol Nrf2. (**B**) Cells were transfected with an ARE-luciferase reporter construct for 16 h before treatment with 0.3, 1, or 5 μM JM17, 0.3 μM RTA-408, or 10 μM DMF. Nrf2 reporter gene activity was measured by ARE luciferase activity level, which was normalized against the β-galactosidase activity level. Data are presented as the mean ±SD of at least three independent experiments. Values from the treated cells were normalized to those of the MJD78 cells treated with vehicle only. * *p* < 0.05, compared to non-treated MJD78 group; ^#^ *p* < 0.05, compared to MJD26 group.

**Figure 5 antioxidants-11-01389-f005:**
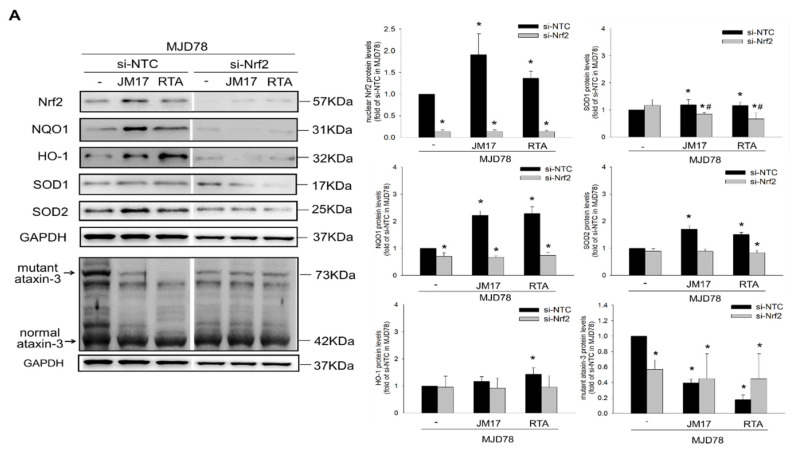
Nrf2 activation-dependent regulation in raising antioxidation and decreasing mutant ataxin-3 expression. MJD78 cells were transiently transfected with si-NTC or si-Nrf2 and with or without an ARE-luciferase reporter construct for 16 h. They were then treated with or without DMSO vehicle, 1 μM JM17 and 0.3 μM RTA-408 for 24 h, respectively. (**A**) Protein expression of Nrf2, NQO1, HO-1, SOD1, SOD2 and mutant and normal ataxin3. Levels of (**B**) ARE luciferase activity, (**C**) total GSH, (**D**) reduced GSH and (**E**) catalase are analyzed after treatments. Data are presented as the mean ±SD of at least three independent experiments. Values from the treated cells were normalized to those of the MJD78 cells treated with si-NTC transfection vehicle only. * *p* < 0.05, compare to non-treated MJD78 with si-NTC group; ^#^ *p* < 0.05, compare to non-treated MJD78 with si-Nrf2 group.

**Figure 6 antioxidants-11-01389-f006:**
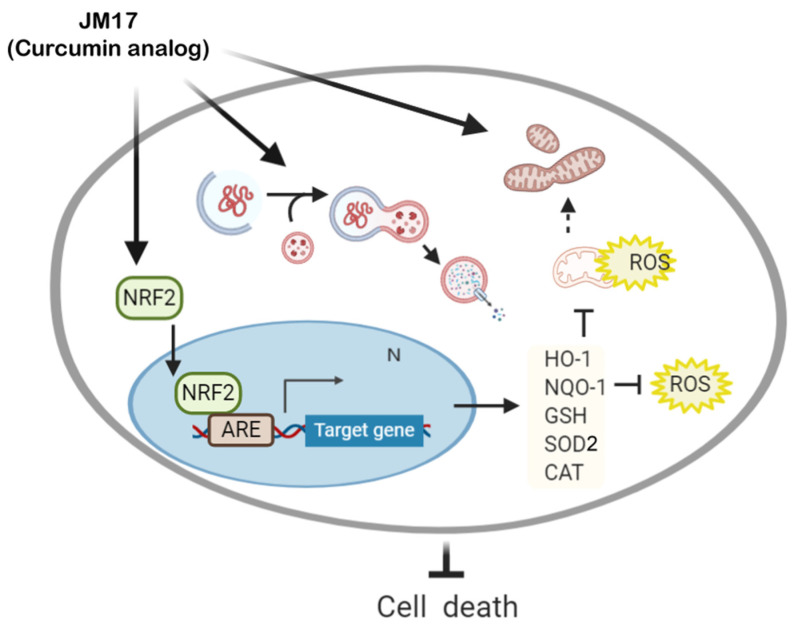
Schematic mechanism of JM17 on Nrf2 activation in MJD78 cells. ROS increased cell death and mitochondrial dysfunction by oxidative stress and ameliorated antioxidant enzyme expression and GSH levels in MJD78 cells expressing mutant ataxin-3 protein. JM17 administration reduced total and mitochondrial ROS and promoted antioxidant enzyme expression by upregulating Nrf2 activation. On the other hand, JM17 diminished protein aggregation by activating autophagy and enhanced mitochondrial respiratory function. NRF2, nuclear factor erythroid-2; ARE, antioxidant response element; ROS, reactive oxygen species; HO-1, heme oxygenase 1; NQO-1, NAD(P)H: quinone oxidoreductase 1; GSH, glutathione; SOD, superoxidase dismutase; CAT, catalase.

## Data Availability

The data presented in this study are available on request from the corresponding author.

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
