# Peer review of "In Vitro Efficacy and Molecular Mechanism of Curcumin Analog in Pathological Regulation of Spinocerebellar Ataxia Type 3"

_antioxidants, 2022, doi:10.3390/antiox11071389_

Round 1
Reviewer 1 Report
The manuscript presented is engaging and helps in giving vital information regarding this rare neurodegenerative disease. The authors have proved their hypothesis with a very large amount of experiments.
Moreover, it is well written and easy to read and the results presented are very interesting and original for the scientific community. In my opinion, this paper can be accepted for publication in "Antioxidants" with minor revision.
I suggest changing the title from "Efficacy and Molecular Mechanism of Curcumin Analog in Pathological Regulation of Spinocerebellar Ataxia Type 3 In Vitro" to "In Vitro Efficacy and Molecular Mechanism of Curcumin Analog in Pathological Regulation of Spinocerebellar Ataxia Type 3".
In 2.4. Mitochondrial respiration paragraph, the authors should replace the 2 of O2 as subscript
The authors shoud increase the quality of figure 2D because it results blurry
In line 476 the authors have to correct the figure number in Figure 6. Regarding this figure there are some parts not described as a central part, please the authors should indicate in the figure what they drew.
Author Response
Point 1: I suggest changing the title from "Efficacy and Molecular Mechanism of Curcumin Analog in Pathological Regulation of Spinocerebellar Ataxia Type 3 In Vitro" to "In Vitro Efficacy and Molecular Mechanism of Curcumin Analog in Pathological Regulation of Spinocerebellar Ataxia Type 3".
Response 1: Thanks for your suggestion and the title has been changed to "In Vitro Efficacy and Molecular Mechanism of Curcumin Analog in Pathological Regulation of Spinocerebellar Ataxia Type 3" in the revision.
Point 2: In 2.4. Mitochondrial respiration paragraph, the authors should replace the 2 of O2 as subscript.
Response 2: Thanks for your detailed correction and it has been corrected in the revision. Please refer to line158,159, 163 and 262 in the revised draft.
Point 3: The authors shoud increase the quality of figure 2D because it results blurry.
Response 3: The figure 2D has been adjusted to make it as clear as possible. I hope it can meet your requirements of image quality.
Point 4: In line 476 the authors have to correct the figure number in Figure 6. Regarding this figure there are some parts not described as a central part, please the authors should indicate in the figure what they drew.
Response 4: The wrong figure number has been corrected (line 522). The simply supplementary description of Figure 6 has been added (line 509-510) in the revision. The detailed description of Figure 6 has been confirmed to explain in the legend. Thanks for your reminding.
"Please see the attachment."

Reviewer 2 Report
The present manuscript “Efficacy and Molecular Mechanism of Curcumin Analog in Pathological Regulation of Spinocerebellar Ataxia Type 3 in Vitro” submitted by Yu-Ling Wu and coworkers reports the in vitro characterization of a curcumin analog (ASC-JM17). In this work, the authors used a cellular model of SK-N-SH cells transfected with ataxin-3 containing 78 CAG repeats to study the effects of the ASC-JM17 compound on oxidative stress, mitochondrial function, and autophagy. Overall, the manuscript is of good quality and the data presented is interesting for the field. However, the manuscript has some important issues that should be addressed before its publication in Antioxidants.
Major comments and suggestions
- Result section. Current manuscript presentation is somehow confusing. Text from section 3.3 ends in line 318. Another formatting issue is that Figure 3 is not cited in the text. Figures 4 and 5 should be presented after they are cited.
- Raw data: Raw blots corresponding to Figure 1C should be also provided.
- Discussion section: The authors claim a major effect of ASC-JM17 on activating the Nrf1/Nrf2 pathway. It would be interesting if they can discuss a possible direct role of ASC-JM17 on inhibiting ataxin-3 aggregation. Previous works demonstrated that curcumin is able to bind and dissociate Tau aggregates (see J Alzheimers Dis. 2017;60(3):999-1014. doi: 10.3233/JAD-170351). This is an important point that should be discussed in order to explain whether the action of ASC-JM17 may be also mediated as a Tau aggregation inhibitor.
Author Response
Point 1: Result section. Current manuscript presentation is somehow confusing. Text from section 3.3 ends in line 318. Another formatting issue is that Figure 3 is not cited in the text. Figures 4 and 5 should be presented after they are cited.
Response 1: Sorry for the formatting dislocation to make you confusion. We have corrected the wrong choreography in section 3.3 of the revision (please refer to line 343-361) as well as the presented order of Figure 3, 4 and 5 (Page 10-13). Thank you for your careful correction.
Point 2: Raw blots corresponding to Figure 1C should be also provided.
Response 2: Figure 1C is the data of respiration rate measured by an Oroboros 2K respirometer and its raw data at three independent experiments (the original typical trace of respirometry measurements recording) has been enclosed in the revised supplementary Figure S4.
Point 3: Discussion section: The authors claim a major effect of ASC-JM17 on activating the Nrf1/Nrf2 pathway. It would be interesting if they can discuss a possible direct role of ASC-JM17 on inhibiting ataxin-3 aggregation. Previous works demonstrated that curcumin is able to bind and dissociate Tau aggregates (see J Alzheimers Dis. 2017;60(3):999-1014. doi: 10.3233/JAD-170351). This is an important point that should be discussed in order to explain whether the action of ASC-JM17 may be also mediated as a Tau aggregation inhibitor.
Response 3: Thanks for your helpful comments to point out another crucial factor of mutant protein formation and dissociation which could be involved in JM17’s regulation. It is clearly deserving further research not only in SCA3 but also in the other polyQ diseases. The reference and related description have been added in the revision (line 448-451).
"Please see the attachment."

Round 2
Reviewer 2 Report
The authors addressed all the issus present in the initial version of the manuscript. The current revised version is suitable for publication.